# Profiling the Major Aroma-Active Compounds of Microwave-Dried Jujube Slices through Molecular Sensory Science Approaches

**DOI:** 10.3390/foods12163012

**Published:** 2023-08-10

**Authors:** Xinhuan Yan, Shaoxiang Pan, Xuemei Liu, Mengnan Tan, Xiaodong Zheng, Wenyu Du, Maoyu Wu, Ye Song

**Affiliations:** 1Jinan Fruit Research Institute, All China Federation of Supply & Marketing Co-Operatives, Jinan 250014, China; yanjuan1206@126.com (X.Y.); panshaoxiang@126.com (S.P.); liuxm0218@163.com (X.L.); mntan28@163.com (M.T.); zhxd1106@163.com (X.Z.); dwy15169137686@163.com (W.D.); wmyu1972@hotmail.com (M.W.); 2Shandong Province Fruit and Vegetable Storage and Processing Technology Innovation Center, Jinan 250014, China

**Keywords:** aroma-active compound, odor activity value, partial least squares regression, dried jujube slices

## Abstract

To discriminate the aroma-active compounds in dried jujube slices through microwave-dried treatments and understand their sensory attributes, odor activity value (OAV) and detection frequency analysis (DFA) combined with sensory analysis and analyzed through partial least squares regression analysis (PLSR) were used collaboratively. A total of 21 major aromatic active compounds were identified, among which 4-hexanolide, 4-cyclopentene-1,3-dione, 5-methyl-2(5H)-furanone, 4-hydroxy-2,5-dimethyl-3(2H)furanone, 3,5-dihydroxy-2-methyl-4-pyrone were first confirmed as aromatic compounds of jujube. Sensory evaluation revealed that the major characteristic aromas of dried jujube slices were caramel flavor, roasted sweet flavor, and bitter and burnt flavors. The PLSR results showed that certain compounds were related to specific taste attributes. 2,3-butanedione and acetoin had a significant positive correlation with the roasted sweet attribute. On the other hand, γ-butyrolactone, 4-cyclopentene-1,3-dione, and 4-hydroxy-2,5-dimethyl-3(2H)furanone had a significant positive impact on the caramel attributes. For the bitter attribute, 2-acetylfuran and 5-methyl-2(5H)-furanone were positively correlated. Regarding the burnt flavor, 5-methyl-2-furancarboxaldehyde and 3,5-dihydroxy-2-methyl-4-pyrone were the most influential odor-active compounds. Finally, 2-furanmethanol and 2,3-dihydro-3,5-dihydroxy-6-methyl-4H-pyran-4-one were identified as the primary sources of the burnt and bitter flavors. Importantly, this work could provide a theoretical basis for aroma control during dried jujube slices processing.

## 1. Introduction

The jujube (*Zizyphus jujuba* Mill., family Rhamnaceae), a traditional food in China, is attractive to consumers due to its rich nutrition, good taste, and potential health benefits [1]. The jujube was harvested from 26,000 hectares of land, and its production was 7.35 million tons in 2022 [2]. Since 2012, there has been an oversupply of jujube on the market because of the excessive production based on large-scale cultivation [3]. Notably, processing is an effective way to reduce post-harvest losses [4]. Among various processed jujube products, dried jujube slices are the most favorable, with the best taste, rich nutritional value, and easy to eat. Additionally, jujube slices provide good economic benefits and broad prospects for development [5]. Dried jujube slices are mostly prepared—including cleaning, coring, and slicing—from naturally dried jujube (about 30% moisture content). The main drying techniques include hot air, microwave, freeze, variable temperature, and pressure differential expansion drying [6]. Microwave drying is a widely applied method in the food industry, with easy operation, low cost, and a short drying cycle. Nowadays, researchers pay more attention to the technical energy consumption, appearance quality, and nutrient retention of different drying processes. Gao et al. [7] investigated the changes in sugars, organic acids, α-tocopherol, β-carotene, phenolic profiles, total phenolic content (TPC), and antioxidant capacity of jujube with four drying treatments (sun, oven, microwave, and freeze drying) [7]. Ji et al. [8] also compared the structural characterization and antioxidant activity of the polysaccharides of jujube (*Zizyphus jujuba* Mill., family Rhamnaceae) [8]. In contrast, research on the aroma of jujube slices has rarely been reported. Notably, the flavor is essential for the quality and acceptability of dried slices [9,10]. After microwave drying, the products could have a caramel and sweet roasted flavor similar to toast or coffee aroma under high temperatures and with enough oxygen. The unique caramel and roasted sweet flavor could benefit the products’ good flavor [11]. However, unpleasant odor, such as burnt and bitter, could be obtained with an unsuitable drying procedure, which has a bad influence on jujube slices’ quality. It is important to study the flavor components and their dynamic characteristics to optimize and control the process parameters during the processing of dried jujube slices.

Recently, some research regarding the aroma of dried jujube was carried out. Chen et al. [12] used headspace solid-phase microextraction gas chromatography–mass spectrometry (HS-SPME/GC-MS) and electronic nose (e-nose) to study a total of 51 aroma compounds from 10 different varieties of dried jujube [1,12]. Additionally, Song investigated the volatile compounds of Chinese jujubes with different drying methods based on metal oxide semiconductor (MOS) e-nose and flash GC e-nose [13]. The above reports illustrated that the drying procedures significantly influenced the variety and amount of volatile compounds and the flavor profile of dried jujube. However, there is a lack of theoretical basis for studying aroma changes in dried jujube during processing. Identifying effective aroma compounds offers a convenient approach to comprehending the sensory alterations, but their research and application in dried jujube slice products are relatively limited.

This study aimed to identify and measure the aroma-active compounds and their dynamics in dried jujube slices using GC-MS combined with GC-O. Additionally, we wanted to evaluate their contribution to the overall aroma by calculating the odor activity value (OAV) and detection frequency analysis (DFA) and to investigate the relationship between sensory properties and characteristic aromatic active compounds using multivariate analysis of partial least squares regression analysis (PLSR).

## 2. Materials and Methods

### 2.1. Raw Material

The jujube in the experiment was ordinary gray jujube purchased from Xinjiang jujube industry Co., Ltd. (Hetian, China) in August 2020 and transported to the laboratory within 72 h at room temperature. After arriving at the laboratory, the water and sugar contents were separately measured according to the method of Gao et al. [6], with 21.0% water content and 69.91% total sugar content.

### 2.2. Preparation of Microwave-Dried Jujube Slices

Before drying, approximately 10 kg of jujube was cut into slices, pitted, and baked in microwave equipment (SANLE, Nanjing, China) with 1 KW power. Seven kinds of jujube slices with different flavor characteristics were prepared by controlling the drying time (0 min, 1 min, 2 min, 2.5 min, 3 min, 4 min, and 5 min). Each drying experiment was carried out in triplicate. They were numbered as B1 (0 min), B2 (1 min), B3 (2 min), B4 (2.5 min), B5 (3 min), B6 (4 min), and B7 (5 min). Each sample was divided into two parts: approximately half was used for the sensory evaluation, and the other was grounded into powder by a food processor (FW100, Taisite Instrument Co., Ltd., Tianjin, China) for the aroma compounds’ analyses. Subsequently, all samples were promptly packed in polyethylene bags under vacuum conditions to mitigate any potential alterations and stored at a temperature of −18 °C in a freezer until further analysis. Jujube powder sample (100 g), deionized water (320 g), 20% sodium chloride solution (48 g), and 1% sodium fluoride solution (32 g) were placed together in a pulper to make a puree. The purpose of adding 1% sodium fluoride solution is mainly to increase ionic strength, which is used to improve sensitivity [14]. Furthermore, the puree samples were immediately used for the analytical determinations with GC usage [14].

### 2.3. Analytical Determinations

#### 2.3.1. Extraction of Volatile Compounds through Drying Jujube with Headspace Solid-Phase Microextraction (HS-SPME)

Microextraction (HS-SPME) was used to extract volatile compounds from dried jujube slices. Non-polar poly dimethy Isiloxane (PDMS) fiber was preferred for the extraction of non-polar analytes. Mixed coating fibers, containing divinylbenzene (DVB) copolymers and carboxen (CAR), could increase retention capacity. PDMS/DVB and CAR/DVB were used for the extraction of low molecular weight volatile and polar analytes [15,16]. The 50/30/30 µm divinylbenzene/carboxen/polydimethylsiloxane (DVB/CAR/PDMS) coated fibers (Supelco, Bellefonte, PA, USA) were used to isolate the volatile compounds once the fibers were conditioned at 250 °C for 30 min. Before the formal experiment, we investigated the effects of the main parameters (including extraction time, sample amount, extraction temperature) on extraction efficiency. Three extraction times (20, 25, and 30 min), three different sample amounts (2, 5, and 8 g), and extraction temperatures (30, 40, and 50 °C) were assayed. The number of compounds and the total peak area were used to evaluate the extraction effect; then, the optimal optimization conditions were determined. For each sample, 5 g of jujube puree was added to a 20 mL headspace bottle (Supelco, Bellefonte, PA, USA) supplemented with 20 μL 2-octanol (32.88 μg/mL in methanol; Sigma-Aldrich, St. Louis, MO, USA) as an internal standard. The headspace bottles were placed in a multifunctional sampler (MPS 2XL; Gerstel GmbH & Co. KG, Mülheim an der Ruhr, Germany) and extracted for 25 min by vibrating at 40 °C. All experiments and sample measurements were carried out in triplicate, and the average values were recorded.

#### 2.3.2. GC-MS Analysis

A GC-MS apparatus (Thermo, Santa Clara, CA, USA) matched with a DB-WAX capillary column (30 mm × 0.25 mm × 0.25 μm) was utilized to isolate and identify volatile compounds in jujube slices. GC-MS (Thermo, Santa Clara, CA, USA) conditions were set as follows: kept at 50 °C for 7 min, ramped at 3 °C/min to 150 °C, increased at 10 °C/min to 250 °C, and maintained for 5 min. Helium, with a purity of 99.999%, served as the carrier gas at a flow rate of 1.5 mL/min in splitless GC inlet mode. MS fragmentation was conducted using electronic impact mode with an ionization energy of 70 eV and a source temperature of 250 °C. Moreover, the transmission line temperature was maintained at 250 °C. Full-scan mode was employed for acquisition, covering a mass range of 50–500 *m*/*z*.

#### 2.3.3. Qualitative and Semiquantitative Analyses

The volatiles were identified based on their mass spectra using the NIST10 library of the GC-MS data system, retention indices (RI) with reference values, and odor descriptions of authentic standards. Additionally, RI values were determined by employing n-alkanes, specifically C7–C30 from Sigma-Aldrich, as standards under the same instrument conditions. A compound was identified if the difference between the calculated and published RI values was less than 20.

With considerations given to cost effectiveness and practicality, an internal standard method was employed to quantify the identified volatiles. This approach aims to establish an efficient means of characterizing the behavior of aroma compounds. The concentration of each identified compound was calculated by comparing its peak area to the peak area of the internal standard (2-octanol). For ease of calculation, a calibration factor of 1.00 was utilized and determined using the following formula, according to the research results of Ref [17]: (1)$$m_{s}=\frac{m_{i}\times A_{S}}{A_{i}\times m_{0}}\times 1000$$

In the formula, *m_s_* represents the concentrations of the identified volatiles in μg/kg; *m_i_* represents the weight of the internal standard in μg; *m*_0_ represents the weight of the jujube slice used in grams; *A_S_* represents the peak area of the identified volatiles; and *A*_i_ represents the peak area of the internal standard.

#### 2.3.4. Calculation of OAVs

The OAV is a metric used to assess the contribution of compounds to the overall aroma profile. It is calculated by dividing the concentration of compounds by their odor threshold in water. The odor threshold represents the minimum concentration humans can perceive in the compound. We referred to previous publications [18] to obtain the necessary data for calculating the OAV. Compounds with an OAV value equal to or greater than 1 are considered potential contributors to the aroma profile of the sample [19,20]. Importantly, these compounds are likely to noticeably impact the overall aroma perception.

#### 2.3.5. Gas Chromatography–Olfactometry (GC-O) Frequency Analysis

To characterize aroma-active compounds, an olfactory detector port (ODP 3; Gerstel GmbH & Co. KG) was employed in conjunction with a GC-MS instrument (Thermo, Santa Clara, CA, USA). This setup facilitated the differentiation of these compounds. After passing through the capillary column, the effluent was evenly split between the sniffing port and the MS detector. The transfer line leading to the GC-O sniffing port was maintained at 220 °C. To prevent nasal dryness, water was added to humidify the effluent, resulting in a flow rate of 60 mL/min. The remaining working conditions were consistent with those previously mentioned for the GC-MS analysis. Ten assessors, consisting of five males and five females, with over 300 h of technical experience, were selected. Prior to olfactory analysis, artificial odor solutions were used for odor identification. Assessors were instructed to record the samples’ retention time and detection frequency (DF). An odorant with a DF (5) could be considered a potential contributor to the aroma profile [1,20].

#### 2.3.6. Sensory Evaluation

Sensory evaluations were conducted to describe the differences in dried jujube slices after undergoing various microwave treatments. The descriptors were selected from the words described by 10 professional reviewers, and words with 80% or more of the votes were counted as evaluation words. Subsequently, organoleptic characteristic descriptors were quantified using five sensory attributes (caramel flavor, roasted sweet flavor, bitter, burnt, and jujube-ID) [5]. The training primarily focused on sensory and semiquantitative descriptive analysis, following the ISO international standard (8586-1, 1993). During the 3-month training period, which consisted of 2 h per week, the panelists were trained to develop the ability to differentiate between different levels of aromas. The subsequent stage of training involved describing and discussing the aroma characteristics of dried jujube slices. The results were modified and annotated in the manuscript.

To evaluate the sensory characteristics of dried jujube slices treated with different microwave power, 5–6 slices of each sample were placed in a 50 mL plastic cup and covered. The cups were prepared 2 h before reaching headspace equilibrium at room temperature and randomly coded with a 3-digit number. Trained assessors used a 10-point scale with 1-unit increments from 0 to 10 to rate each descriptor, and there was a 3 min interval between each sample evaluation. Chinese panelists commonly use this scale due to its intuitive and easy-to-understand nature. Furthermore, the scale is broad enough to encompass the full sensory attribute intensities. It also has sufficient discrete points to distinguish subtle differences in intensity between samples [21,22]. The three independent QDA tests for each odor descriptor were averaged and plotted in a spider web chart.

### 2.4. Statistical Analysis

The experiments were conducted in triplicate, and the obtained data were analyzed using ANOVA. Duncan’s multiple range test (*p* < 0.05) was employed to assess the differences. The statistical analysis was conducted using the SPSS software (SPSS Statistics 19.0; IBM Corporation, New York, NY, USA). Cluster analysis was accomplished using Rstudio (R-Tools Technology, Inc., Richmond Hill, ON, Canada).

The Unscrambler software version 9.7 (CAMO ASA, Oslo, Norway) was utilized to analyze the correlations between sensory attributes and volatile compounds and to generate the PLSR plot.

## 3. Results and Discussion

### 3.1. Profile of Dried Jujube Slices

Among the seven jujube slice samples, there were significant differences in the quantity and composition of the volatiles. A total of 83 volatile compounds were detected in the 7 samples, including 12 aldehydes, 21 esters, 18 ketones, 9 alcohols, 15 acids, 6 hydrocarbons, and 2 furans (Table 1). Most of the volatile compounds detected in this study agreed with previous research on dried jujube slices [23]. Compared with Ref [23], some new compounds were found, namely 4-hexanolide, 4-cyclopentene-1,3-dione, 5-methyl-2(5H)-furanone, 4-hydroxy-2,5-dimethyl-3(2H)furanone, and 3,5-dihydroxy-2-methyl-4-pyrone. Among the volatiles, acids accounted for the highest semiquantitative volatile portion (Figure 1), and 15 acids were identified in all samples. 4-Methylvaleric acid, trans-3-hexenoic acid, and decanoic acid were only detected in the B1 sample. The second highest semiquantitative volatile portion consisted of ketones in the B2–B7 samples (Figure 1). A total of 18 ketones were identified, the main components of which were furanone and pyranone. About 21 esters were identified during drying, regarded as one of the most important flavoring compounds in fresh jujube [24]. The most quantity-predominant compound in dried jujube slices was acetic acid (581.6 ug/kg in the B1 sample) (61 in Table 1). The acetic acid content in this study was consistent with previously reported quantification studies of jujube [25].

### 3.2. Identification of Aroma-Active Compounds by OAV

The OAVs have been widely used to assess the aroma potency of foods by considering the balance between the food matrix and the surrounding air, as reported in previous studies [28,29]. In this study, the OAVs of 83 volatile compounds with documented odor threshold values, as shown in Table 1, were calculated to evaluate their contribution to the overall aroma of dried jujube slices [30]. Thirty-one volatile compounds were excluded from the analysis due to insufficient odor threshold data. Among the remaining compounds, 17 were identified as aroma-active based on their OAV values, with a relatively high contribution to the overall aroma. The OAV contribution was dominated by esters (94.03%) in the B1 sample, with hexyl acetate, which had a strong fruity aroma, having the highest OAV of 278.48. Ketones (33.88–42.86%) dominated their OAV contribution in B3–B6 samples, three of which, with the strong roasted sweet-like 2,3-Butanedione aroma, had the highest OAV of 2.3–12.7. Furthermore, furanones and pyranones, typically produced by lipid peroxidation and Maillard reactions of carbonyl and amino compounds, can significantly impact the aroma profile of food [31]. However, in identifying aroma-active compounds, it is also essential to consider detection frequency analysis findings.

### 3.3. Identification of Aroma-Active Compounds by DFA

DFA is a common method for identifying aroma compounds, as it allows for characterizing compounds, which contribute to the overall aroma perception [32]. To be considered an aroma-active compound, it must be detected by at least 5 out of the 10 panelists in the discrimination test.

Lina Wang [33] used “roast”, “sweet”, “green”, “sour”, “fruity”, and “mellow” attributes to describe the odor of jujubes. Moreover, bitter and jujube-ID were evaluated as important attributes in analyzing the flavor of Spanish jujube (*Zizyphus jujuba* Mill., family Rhamnaceae) fruits [34]. Notably, different substances are present in different odors. A total of 15 aromatic aroma compounds were identified by matching the LRI and mass spectrum. These compounds were then categorized into five groups according to their odor characteristics. These were roasted sweet odorants (34 and 35 in Table 1), caramel-like odorants (26 and 38 in Table 1), burnt-like odorants (10, 12, 49, 50, and 57 in Table 1), bitter-like odorants (40, 49, 50, and 82 in Table 1), and jujube-ID odorants (13, 27, 52, and 67 in Table 1). The jujube-ID odorants consisted of mushroom, smelly, sour, and sweet substances.

### 3.4. Comparison of DFA and OAV Aroma-Active Compounds’ Identification

Eleven volatile compounds were identified as potential aroma-active compounds in dried jujube slices based on the joint analysis of OAV and DFA, such as 4-hexanolide, 5-methyl-2(5H)-furanone, 3,5-dihydroxy-2-methyl-4-pyrone, 4-hydroxy-2,5-dimethyl-3(2H)furanone, 4-cyclopentene-1,3-dione, and so on. Some compounds with high DF could not be detected by OAV, such as 2,3-butanedione and acetoin in the B7 sample, which were significantly and positively correlated with roasted sweet attributes. Moreover, some compounds with high OAV could not be recognized by DFA, such as butanoic acid with descriptors of smelly. In food matrices, the release of aroma compounds can be influenced by interactions between the volatiles and the components of the food [1]. Thus, we should use both methods to identify aromatic active compounds. The results of OAV were in good agreement with DFA in identifying the major contributors to the aroma of dried jujube slices. Furthermore, 21 compounds were recognized as significant aroma-active compounds in this study. The cluster heat map of the 21 compounds of the samples under different treatment conditions is shown in Figure 2.

### 3.5. Effect of Different Microwave Drying Times on the Major Aroma-Active Compounds in Jujube

To characterize the aroma profile of jujube slices dried with microwave treatments, major aroma-active compounds (21 in total) were identified through OAV and DFA, and the results are presented in Table 2. Hierarchical clustering, including heat maps (Figure 2), was performed based on the changes in the identified aroma compounds to illustrate the aroma profile of the seven jujube slice samples with different processing methods. Cluster analysis indicated that seven samples were distinguished into three classes (B1 and B2, B3 to B6, B7). Moreover, the changes in major aroma-active compounds and the widening disparities from B1 to B7 with the extension of drying time indicated that the characteristic aroma changed the processing.

Esters are regarded as one of jujube’s most important flavoring compounds [24]. Among the three aroma-active esters, ethyl hexanoate (fruity) was found to decline significantly with the increase in drying time, and it was speculated that an increase in the heat treatment time could promote the degradation of ethyl hexanoate, which agrees with the findings of previous studies.

5-Acetyldihydrofuran-2(3H)-one (sweet) was first discovered in jujube, and the variation tendency agreed with ethyl hexanoate. As for the microwave treatments, the concentrations of γ-butyrolactone tended to rise first and then decrease along with the increase in drying time. One possible explanation for this effect is the inactivation of ester synthase and the Maillard reaction of valine and isoleucine, which only occurs within specific drying time ranges. However, additional research is needed to confirm this proposed explanation.

Five aroma-active aldehydes were identified in jujube slices, including hexanal and nonanal. These compounds are commonly associated with “green”, “cut grass”, “fat”, and “citrus” notes, which were also noted by panelists in the sensory descriptions of jujube slices and are typical of many other jujube varieties [3,35,36]. Hexanal and nonanal showed a decreasing tendency in response to drying time, possibly due to thermal degradation and volatilization loss. Furfural, 5-methyl-2-furanaldehyde, and 5-hydroxymethyl-2-furaldehyde were not detected in the fresh jujube and significantly increased (*p* < 0.05) with the increase in drying time. It is well known that furfural and furan aldehydes are formed by the Maillard reaction of pentose or hexose [37,38], contributing to burnt attributes. Based on this theory, the laws of generation and change of these three aldehydes can be well explained.

Ketones and furans were the most diverse components among the major aroma-active compounds of jujube slices. In contrast, seven aroma-impacting ketones and one furan were found in the jujube samples. They were 2,3-butanedione (roasted sweet), acetoin (roasted sweet), 4-hydroxy-2,5-dimethyl-3(2H)furanone (caramel), 4-cyclopentene-1,3-dione (caramel), 5-methyl-2(5H)-furanone (bitter), 2,3-dihydro-3,5-dihydroxy-6-methyl-4(H)-pyran-4-one (burnt, bitter), 2-acetylfuran (bitter), 3,5-dihydroxy-2-methyl-4-pyrone (burnt), which were formed during heating, except 4-cyclopentene-1,3-dione. The results of olfactometric analysis and compound flavor description showed that ketones and furans were important parts of the flavor of dried jujube slices. Studies have also shown that the formation of ketones and furans was related to the Maillard reaction, Strecker degradation reaction, and secondary reaction of fat thermal oxidation and thermal degradation [24]. The concentrations of the three ketones 4-hydroxy-2,5-dimethyl-3(2H)furanone, 4-cyclopentene-1,3-dione, and acetoin(2,3-butanedione) decreased, starting with sample B5. This may be because the intermediate ketones continue to react with amino acids to produce melaniginoids with the intensification of the Maillard reaction. However, the pattern of change in ketones requires further study and discussion.

Two alcohols (1-octen-3-ol, 2-furanmethanol) and three acids (acetic acid, butanoic acid, hexanoic acid) were identified as aroma-active compounds of jujube, which were found to decrease after microwave treatments, except for 2-furanmethanol. 2-Furanmethanol (burnt and bitter flavor) was the typical product of the Maillard reaction [25], and it was one of the typical flavor substances of dried jujube. With a critical contribution to the flavor of jujube slice samples, the typical C8 alcohol, 1-octen-3-ol, with a mushroom note (also known as “mushroom alcohol”), was detected. Additionally, its concentration decreased with increasing drying time, which may contribute to reducing the floral attribute in jujube slice samples. Acids were identified as the key aroma contributor in jujube samples [36,39]. According to the sensory results, the sour taste was one of the important flavor characteristics of the original jujube slices. Notably, the sour characteristic gradually decreased with the increase in drying time. This change in sensory characteristics was consistent with the trend of acid content. Furthermore, the acid decrease was probably due to the formation of esters and volatilization loss during the drying process [25].

### 3.6. Sensory Analysis

The results of the sensory evaluation analysis of dried jujube slices are shown in the radar charts in Figure 3. In addition to jujube-ID, roasted sweet, caramel, bitter, and burnt were the primary characteristic flavors of jujube slices formed during drying. These flavors together constituted the unique flavor of red jujube after baking. Additionally, the drying time significantly affected the intensities of the basic tastes (roasted sweet, caramel, bitter, and burnt). With the increase in drying time, roasted sweet and caramel increased first and then decreased, and the bitter and burnt flavor increased. For instance, the scores of the B4 sample were the highest in roasted sweet (9.0) and caramel (8.0) attributes. Both 2,3-butanedione and acetoin showed high content and a roasted sweet smell contributing to the “roasted sweet” and “caramel” attributes of the B4 sample. The B7 sample had the highest rated value of the “bitter” and “burnt” descriptors, while the B1 sample showed the lowest sensorial score. This is likely because the adjustment of the treatment drying time resulted in significantly high intensities of roasted sweet and caramel flavors, which was linked to intense drying characteristic aroma. On the other hand, it produced burnt–bitter notes due to excessive heating. At the same time, the range of intensity of the jujube-ID as the important flavor parameter, which represented the flavor of dried jujube, was 9.5–1.0, with a mean value of 4.5; the highest values of this essential attribute were found in dried jujube, and the values then decreased with increasing the time of drying. For instance, samples B6 (30 min) and B7 (45 min) had values of 2.0 and 1.0, respectively. The B1 samples were described more often as having a jujube-ID flavor than the other samples, and acids could perhaps account for the similarity in jujube-ID. Based on the radar charts, the B4 samples were considered caramel and roasted sweet. The overall profiles of the B7 samples were small in the radar charts, given that they were described as bitter and burnt with weak caramel and roasted sweet flavors. The B7 sample showed a strongly bitter note compared to other dried jujube slice samples, mainly because of the high content of 2-acetylfuran (22.40 μg/kg). The different kinds, relative contents, sensory threshold values of aroma-active compounds, and the mutual synergism among aroma-active compounds determined the sensory characteristics of jujube slices. Furthermore, the sensory analysis also agrees with the results of GC-MS and GC-O.

### 3.7. PLSR

To investigate the correlation between the major aroma-active compounds and sensory attributes of dried jujube slices, a PLSR analysis was carried out. The selection of compounds for this analysis is typically based on their relevance and significance in relation to the research objectives. In this case, the 21 aroma-active compounds identified through GC-MS detection and subsequent OAV and DFA analysis were chosen as the predictor variables for the PLSR analysis. These compounds were considered important in terms of their potential contribution to the aroma profile of jujube slices. The 21 aroma-active compounds identified in the jujube slices (Table 2) were subjected to PLSR using PLS1 and PLS2 methods. The X-matrix represented the aroma-active compounds, while the Y-matrix represented one or more sensory attributes. The PLSR2 model extracted two principal components, explaining 97% of cross-validation variance in the X-variables and 92% in the Y-variables. The resulting correlation loading plot is shown in Figure 4, with the inner and outer ellipses indicating 50% and 100% of the explained variance, respectively. All compounds, seven samples, and five sensory attributes were located between the inner and outer ellipses, with r2 values ranging from 0.5 to 1.0, indicating good agreement with the PLSR model. Furthermore, the 21 major aroma-active compounds significantly impacted one or more of the five sensory descriptors.

PLSR1 regression analysis was conducted to identify the odor-active compounds, which significantly contributed to the important sensory attributes of jujube slices, including jujube-ID, caramel, roasted sweet, bitter, and burnt. The results showed that butanoic acid (63), acetic acid (61), hexanoic acid (67), hexanal (2), and nonanal (5) were positively connected with the jujube-ID attribute (Figure 5a). At the same time, 2,3-butanedione (34) and acetoin (35) were positively related to the roasted sweet attribute (Figure 5c).

Moreover, the caramel attribute was significantly positively correlated with γ-butyrolactone (26), 4-cyclopentene-1,3-dione (38), and 4-hydroxy-2,5-dimethyl-3(2H)furanone (45) (Figure 5b). In addition, 2-acetylfuran (82) and 5-methyl-2(5H)-furanone (40) were positively correlated with the bitter attributes (Figure 5d). At the same time, 5-methyl-2-furanaldehyde (10), furfural (7), and 3,5-dihydroxy-2-methyl-4-pyrone (50) also showed a strong correlation with the burnt attribute (Figure 5e). 2-Furanmethanol (57) and 2,3-dihydro-3,5-dihydroxy-6-methyl-4(H)-pyran-4-one (49) had a significant positive impact on the bitter and burnt attribute, which was consistent with findings by Qiao et al. [40]. Moreover, this result was consistent with our previous sensory evaluation, which found that the flavor of caramel and roasted sweet in the samples first increased rapidly and then decreased. At the same time, the bitter and burnt attributes considerably increased, particularly in B6 and B7 samples. The sensory data results also showed good agreement with the twenty-one aroma-active compounds. The sensory differences and similarities in the flavor of samples were mainly caused by the aroma-active compounds’ composition, content, and OAV [1].

## 4. Conclusions

Through GC-MS detection combined with OAV and DFA analysis methods, we identified the 21 most significant aroma-active compounds in jujube slices, including 17 compounds with OAVs ≥ 1, 15 compounds with DFs ≥ 5, and 11 compounds with both OAVs ≥ 1 and DFs ≥ 5, such as caramel-like 4-cyclopentene-1,3-dione and bitter-like 5-methyl-2(5H)-furanone. Simultaneously, some new compounds were found to possess aroma activity in jujube, namely 5-ethyldihydro-2(3H)-furanone,5-methyl-2(5H)-furanone, 3-Hydroxy-2-methyl-4H-pyran-4-one, 2,5-Dimethyl-4-hydroxy-3(2H)-furanone, and 4-Cyclopentene-1,3-dione. Furthermore, the PLSR analysis of 21 aroma-active compounds and sensory evaluation results showed that 2,3-butanedione and acetoin were significantly and positively correlated with the roasted sweet attribute, while γ-butyrolactone, 4-cyclopentene-1, 3-dione, and 2,5-dimethyl-4-hydroxy-3(2H)-furanone provided an intense caramel flavor. Moreover, Ethanone, 1-(2-furanyl)- and 5-methyl-2(5H)-furanone were the primary sources of the bitter taste. Notably, the burnt taste in jujube slices was found to be mainly caused by 3-Hydroxy-2-methyl-4H-pyran-4-one, while 2-Furanmethanol and 2,3-dihydro-3,5-dihydroxy-6-methyl-4H-Pyran-4-one were identified as the primary sources of both burnt and bitter flavors. 

Among the 21 flavor compounds, 5-methyl-2(5H)-furanone, 4H-Pyran-4-one, and 2,3-dihydro-3,5-dihydroxy-6-methyl-4H-Pyran-4-one were undetected in the unheated sample (B1). 5-methyl-2(5H)-furanone was also not detected in the sample microwaved for 1 min (B2). Compounds such as 1-(2-furanyl)-ethanone, 5-methyl-2(5H)-furanone, 3-Hydroxy-2-methyl-4H-pyran-4-one, 2-furanmethanol, and 2,3-dihydro-3,5-dihydroxy-6-methyl-4H-pyran-4-one, which exhibit burnt and bitter flavor, gradually increased in content with the increase in microwave time. The contents of compounds (2,3-butanedione, acetoin, γ-butyrolactone, 4-cyclopentene-1,3-dione, and 2,5-dimethyl-4-hydroxy-3(2H)-furanone) exhibiting roasted sweet and caramel flavors initially increased, then decreased, and reached their highest levels at 2.5 min (B4). Importantly, these conclusions could contribute to understanding the material basis of dried jujube slices’ aroma and their change regularities and provide a theoretical basis for improving the aroma quality and regulating peculiar flavor to produce high-quality jujube slices.

## Figures and Tables

**Figure 1 foods-12-03012-f001:**
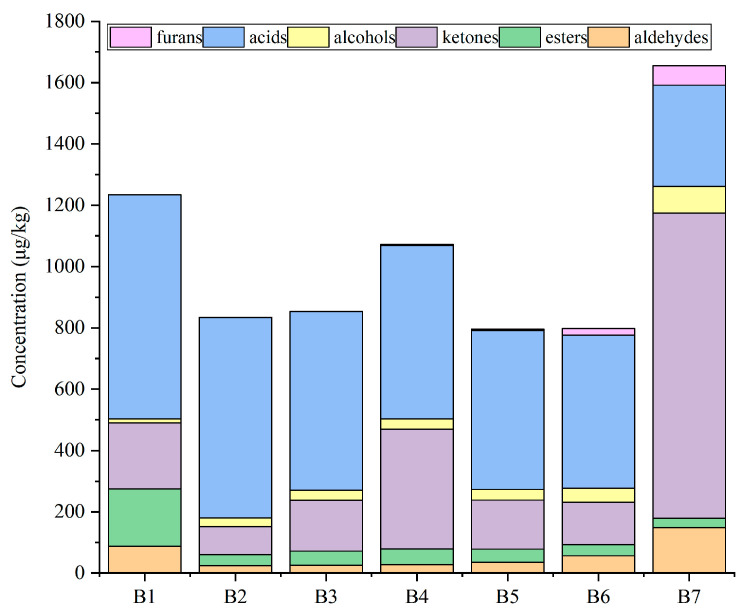
Effect of different microwave drying times on flavor and composition content of jujube slices: (B1) drying time 0 min; (B2) drying time 1 min; (B3) drying time 2 min; (B4) drying time 2.5 min; (B5) drying time 3 min; (B6) drying time 4 min; (B7) drying time 5 min.

**Figure 2 foods-12-03012-f002:**
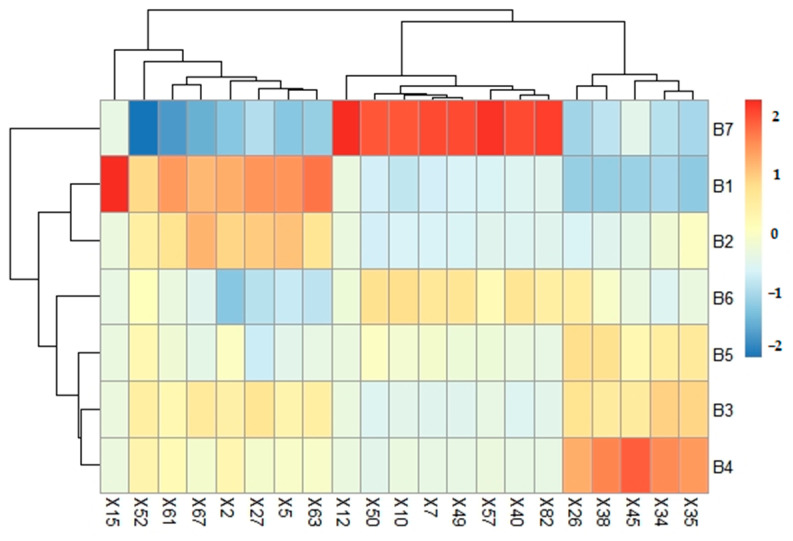
Hierarchical clustering of the 21 major aroma-active compounds in samples with different microwave drying times: (B1) drying time 0 min; (B2) drying time 1 min; (B3) drying time 2 min; (B4) drying time 2.5 min; (B5) drying time 3 min; (B6) drying time 4 min; (B7) drying time 5 min.

**Figure 3 foods-12-03012-f003:**
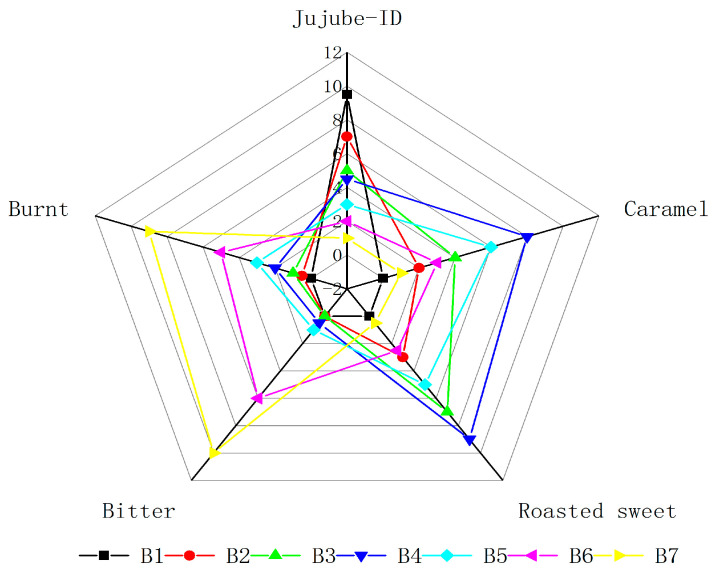
Spider chart of sensory evaluation for jujube slices at different microwave drying times: (B1) drying time 0 min; (B2) drying time 1 min; (B3) drying time 2 min; (B4) drying time 2.5 min; (B5) drying time 3 min; (B6) drying time 4 min; (B7) drying time 5 min.

**Figure 4 foods-12-03012-f004:**
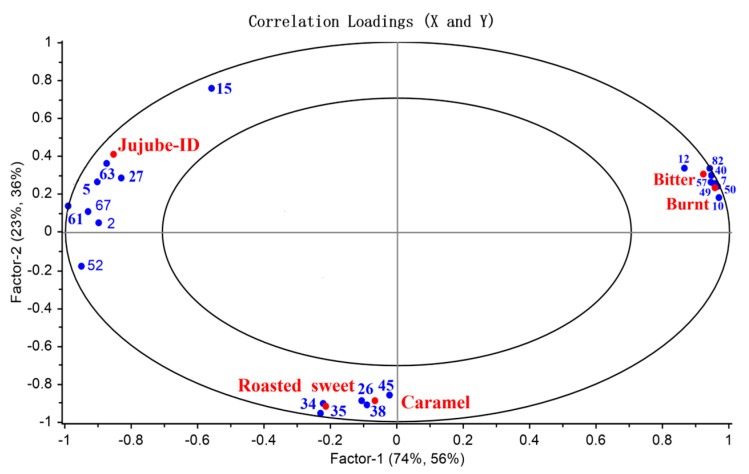
Correlation loadings for odor-active compounds with OAVs > 1 (21 compounds in Table 2; the numbering is consistent with the compound numbering in Table 2) and sensory data (Y-matrix) of all samples by PLSR2.

**Figure 5 foods-12-03012-f005:**
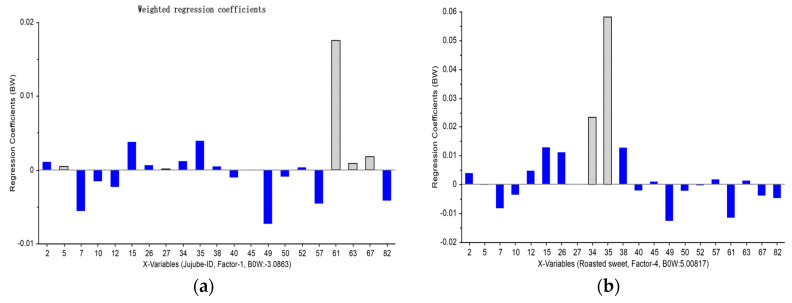
Regression coefficients and significance indications for sensory attributes by PLSR1. (**a**) Jujube factor; (**b**) Caramel factor; (**c**) Roasted sweet factor; (**d**) Bitter factor; (**e**) Burnt factor. Gray columns are the key compounds identified by OAV and DFA.

**Table 1 foods-12-03012-t001:** Volatile compounds’ concentration (μg/kg) and their odor activity values (OAVs) in seven jujube slices.

No	Compound Name ^A^	CAS	RI ^B^	LRI ^C^	Identification ^D^	Concentration (μg kg^−1^)							OT ^F^
						B1 ^I^	B2	B3	B4	B5	B6	B7	/
aldehydes													
1	2-Butenal	4170-30-3	1051	- ^E^	MS,RI	42.9 ± 5.78	0	0	0	0	0	0	/ ^H^
2	Hexanal	66-25-1	1089	1081	MS,RI	15.39 ± 0.65 a ^G^	13.43 ± 0.23 b	10.37 ± 0.31 c	9.52 ± 1.12 d	7.97 ± 0.74 e	0	0	0.0036
3	2-Ethyl-2-butena	19780-25-7	1158	-	MS,RI	3.64 ± 0.52	0	0	0	0	0	0	/
4	(Z)-2-Heptenal	57266-86-1	1271	1291	MS,RI	3.70 ± 0.47	0	0	0	0	0	0	0.01
5	Nonanal	124-19-6	1403	1392	MS,RI	8.38 ± 0.34 a	7.34 ± 0.41 b	5.32 ± 0.54 c	4.23 ± 0.42 d	3.11 ± 0.37 e	2.26 ± 0.53 f	1.02 ± 0.62 g	0.0035
6	(E,E)-2,4-Hexadienal	142-83-6	1418	1403	MS,RI	1.17 ± 0.15	0	0	0	0	0	0	1.6
7	Furfural	98-01-1	1478	1466	MS,RI	0	0.93 ± 0.23 f	5.38 ± 0.78 e	8.34 ± 0.56 d	17.04 ± 0.42 c	38.97 ± 0.34 b	82.5 ± 1.57 a	0.008
8	Decanal	112-31-2	1508	1495	MS,RI	0	0.81 ± 0.11 a	0.97 ± 0.09 a	0.96 ± 0.07 a	0.82 ± 0.15 a	0.51 ± 0.21 b	0	0.0009
9	Benzaldehyde	100-52-7	1541	1538	MS,RI	13.09 ± 1.12	0	0	0	0	0	0	0.05
10	5-Methyl-2-furanaldehyde	620-02-0	1588	1570	MS,RI	0	1.56 ± 0.12 f	3.03 ± 0.21 e	3.87 ± 0.05 d	5.34 ± 0.08 c	13.82 ± 0.11 b	23.19 ± 0.08 a	0.005
11	5-Acetoxymethyl-2-furaldehyde	10551-58-3	2203	-	MS,RI	0	0	0	0	0	0	2.33 ± 0.28	/
12	5-Hydroxymethyl-2-furaldehyde	67-47-0	2514	2510	MS,RI	0	0	0.20 ± 0.04 d	0.42 ± 0.05 c	0.49 ± 0.09 c	1.30 ± 0.25 b	39.3 ± 2.27 a	5
esters													
13	Ethyl acetate	108-05-4	989	-	MS,RI	0	14.4 ± 0.12 a	6.76 ± 0.25 b	4.20 ± 0.13 c	4.76 ± 0.24 c	3.58 ± 0.22 d	1.02 ± 0.09 e	/
14	Ethyl valerate	539-82-2	1141	-	MS,RI	12.2 ± 0.14	0	0	0	0	0	0	0.094
15	Ethyl hexanoate	123-66-0	1240	1238	MS,RI	139.24 ± 5.67 a	4.04 ± 0.21 b	3.49 ± 0.15 b	3.50 ± 0.22 b	2.11 ± 0.13 c	0	0	0.0005
16	Ethyl heptanoate	106-30-9	1340	-	MS,RI	10.04 ± 0.52	0	0	0	0	0	0	0.17
17	2-Hexenoic acid ethyl ester	1552-67-6	1353	-	MS,RI	3.82 ± 0.21	0	0	0	0	0	0	/
18	2,3-Dihydro-5-methyl-2-furanone	591-12-8	1418	-	MS,RI	0	0.53 ± 0.04 c	0.76 ± 0.06 bc	0.84 ± 0.06 b	0.86 ± 0.09 b	0.89 ± 0.12 b	8.92 ± 0.82 a	/
19	Ethyl caprylate	106-32-1	1441	1466	MS,RI	6.53 ± 0.32	0	0	0	0	0	0	0.0001
20	(E)-9-Tetradecen-1-olacetate	23192-82-7	1478	-	MS,RI	0.91 ± 0.08	0	0	0	0	0	0	/
21	Formic acid furfuryl ester	13493-97-5	1507	1481	MS,RI	0	0	0	0	0	0	2.44 ± 0.12	/
22	Methyl decanoate	110-42-9	1601	1590	MS,RI	0.44 ± 0.08	0	0	0	0	0	0	1
23	4-Hydroxy-2-methylbutanoic acid lactone	1679-47-6	1610	-	MS,RI	0.76 ± 0.11 b	1.21 ± 0.14 a	0.95 ± 0.08 b	0.92 ± 0.06 b	0.89 ± 0.06 b	0.86 ± 0.10 b	0	/
24	γ-Valerolactone	108-29-2	1630	-	MS,RI	0	0.56 ± 0.05 a	0.50 ± 0.04 ab	0.44 ± 0.04 b	0.41 ± 0.03 b	0.33 ± 0.04 c	0.30 ± 0.03 c	100
25	Ethyl caprate	110-38-3	1644	-	MS,RI	4.25 ± 0.27	0	0	0	0	0	0	0.02
26	γ-Butyrolactone	96-48-0	1650	-	MS,RI	1.34 ± 0.25 g	10.05 ± 0.42 e	28.80 ± 0.52 c	36.65 ± 0.37 a	30.24 ± 0.48 b	25.64 ± 0.62 d	2.55 ± 0.44 f	0.025
27	4-Hexanolide	695-06-7	1722	-	MS,RI	4.04 ± 0.27 a	3.71 ± 0.36 a	3.50 ± 0.52 ab	2.97 ± 0.21 b	2.53 ± 0.35 b	2.43 ± 0.22 b	2.37 ± 0.19 b	8
28	δ-Hexanolactone	823-22-3	1815	-	MS,RI	0.76 ± 0.05 d	1.52 ± 0.12 a	1.51 ± 0.11 a	1.33 ± 0.13 ab	1.24 ± 0.08 b	1.20 ± 0.07 b	0.93 ± 0.05 c	230
29	γ-Heptanolactone	105-21-5	1825	-	MS,RI	0.60 ± 0.05	0	0	0	0	0	0	0.52
30	Ethyl laurate	106-33-2	1849	-	MS,RI	0.73 ± 0.04	0	0	0	0	0	0	3.5
31	γ-Octanoic lactone	104-50-7	1935	1883	MS,RI	0.55 ± 0.06 b	0.52 ± 0.05 b	0.69 ± 0.04 a	0.59 ± 0.04 ab	0.55 ± 0.06 b	0.67 ± 0.06 a	0.56 ± 0.03 b	0.095
32	Methyl pyruvate	600-22-6	2357	-	MS,RI	0	0	0	0	0	0	10.39 ± 1.24	/
33	3,4-Dihydroxybutanoic acid gamma-lactone	5469-16-9	2618	-	MS,RI	0	0	0	0	0	0	0.92 ± 0.08	/
ketones													
34	2,3-Butanedione	431-03-8	1022	981	MS,RI	0	12.00 ± 2.58 d	29.00 ± 3.12 b	38.00 ± 1.11 a	22.00 ± 0.85 c	7.00 ± 0.41 e	2.00 ± 0.34 f	0.003
35	Acetoin	513-86-0	1294	1292	MS,RI	0	60.28 ± 4.28 d	102.09 ± 3.78 b	128.23 ± 4.12 a	87.45 ± 3.56 c	42.16 ± 2.87 e	8.25 ± 1.56 f	0.04
36	6-Methyl-5-hepten-2-one	110-93-0	1347	1339	MS,RI	0	1.49 ± 0.22 b	1.74 ± 0.15 b	1.81 ± 0.2 ab	1.84 ± 0.34 ab	2.36 ± 0.45 a	2.50 ± 0.52 a	0.1
37	3-Acetoxy-2-butanone	4906-24-5	1390	-	MS,RI	3.18 ± 0.12 d	6.46 ± 0.23 a	5.59 ± 0.18 b	4.41 ± 0.55 c	4.38 ± 0.42 c	3.64 ± 0.43 d	2.08 ± 0.15 e	/
38	4-Cyclopentene-1,3-dione	930-60-9	1602	-	MS,RI	2.12 ± 0.15 g	8.56 ± 0.05 e	18.24 ± 0.07 c	28.37 ± 0.07 a	20.54 ± 0.06 b	12.65 ± 0.08 d	5.12 ± 0.14 f	0.02
39	5,5-Dimethylfuran-2(5H)-one	20019-64-1	1626	-	MS,RI	0	0.42 ± 0.05 a	0.44 ± 0.04 a	0.45 ± 0.06 a	0.46 ± 0.02 a	0.47 ± 0.04 a	0.50 ± 0.03 a	/
40	5-Methyl-2(5H)-furanone	591-11-7	1698	-	MS,RI	0	0	0.01 ± 0.002 d	0.97 ± 0.05 c	1.26 ± 0.04 bc	6.99 ± 0.05 b	15.02 ± 0.15 a	0.002
41	2(5H)-Furanone	497-23-4	1776	-	MS,RI	0	0	0.24 ± 0.03 d	0.25 ± 0.04 cd	0.32 ± 0.04 c	0.48 ± 0.05 b	2.66 ± 0.21 a	/
42	1,2-Cyclopentanedione	3008-40-0	1783	-	MS,RI	0	0	0	0	0	0	1.03 ± 0.11	/
43	3-Methyl-1,2-cyclopentanedione	765-70-8	1842	-	MS,RI	0	0.61 ± 0.08 c	0.95 ± 0.07 b	0.98 ± 0.12 b	1.00 ± 0.09 b	1.07 ± 0.13 ab	1.42 ± 0.24 a	0.01
44	Furyl hydroxymethyl ketone	17678-19-2	2023	-	MS,RI	0	0	0	0	0	0	1.03 ± 0.12	1
45	4-Hydroxy-2,5-dimethyl-3(2H)furanone	3658-77-3	2038	2002	MS,RI	0	0.41 ± 0.09 c	0.96 ± 0.24 b	1.75 ± 0.42 a	0.79 ± 0.15 b	0.46 ± 0.07 c	0.38 ± 0.04 cd	0.001
46	5-Acetyldihydrofuran-2(3H)-one	29393-32-6	2080	-	MS,RI	0	0	0	0	0	0.42 ± 0.05 b	2.13 ± 0.12 a	/
47	2,5-Hexanedione	110-13-4	2120	-	MS,RI	0	0	0	0	0	0	0.60 ± 0.05	/
48	2-Hydroxy-gamma-butyrolactone	19444-84-9	2188	-	MS,RI	0	0	0	0	0	0.43 ± 0.03 b	3.99 ± 0.11a	/
49	2,3-Dihydro-3,5-dihydroxy-6-methyl-4(H)-pyran-4-one	28564-83-2	2286	2266	MS,RI	0	1.13 ± 0.12 f	5.92 ± 0.24 e	11.57 ± 0.21 d	16.50 ± 0.32 c	53.04 ± 0.27 b	106.79 ± 1.24 a	0.02
50	3,5-Dihydroxy-2-methyl-4-pyrone	1073-96-7	2314	-	MS,RI	0	0	0.54	1.12	3.56	7.35	13.12 ± 0.08	0.01
51	Methyl pyruvate	600-22-6	2357	-	MS,RI	0	0	0	0	0	0	10.39 ± 0.85	/
alcohols													
52	1-Octen-3-ol	3391-86-4	1453	1441	MS,RI	5.69 ± 0.52 a	4.87 ± 0.37 bc	4.92 ± 0.35 b	4.69 ± 0.41 bc	4.52 ± 0.25 c	4.23 ± 0.36 c	0	0.001
53	2-Ethylhexanol	104-76-7	1493	-	MS,RI	6.87 ± 0.24 a	4.58 ± 0.35 c	4.73 ± 0.24 c	4.84 ± 0.52 bc	4.92 ± 0.32 bc	5.35 ± 0.29 b	7.20 ± 0.54 a	/
54	(S,S)-2,3-Butanediol	19132-06-0	1543	-	MS,RI	0	13.30 ± 0.55 a	12.42 ± 0.47 b	12.01 ± 0.85 bc	11.72 ± 0.65 c	10.59 ± 0.74 c	0	400
55	Propylene glycol	57-55-6	1595	-	MS,RI	0.83 ± 0.08 c	1.66 ± 0.23 a	1.47 ± 0.21 ab	1.45 ± 0.16 ab	1.35 ± 0.12 b	1.31 ± 0.17 b	1.24 ± 016 b	1400
56	4-Methyl-5-decanol	213547-15-0	1659	-	MS,RI	0.38 ± 0.05	0	0	0	0	0	0	/
57	2-Furanmethanol	98-00-0	1669	-	MS,RI	0	1.49 ± 0.18 f	5.57 ± 0.24 e	7.23 ± 0.32 d	8.78 ± 0.34 c	20.44 ± 1.24 b	73.57 ± 3.25 a	0.3
58	5-Methyl-2-furanMethanol	3857-25-8	1728	-	MS,RI	0	0	0.64 ± 0.07 c	0.75 ± 0.09 c	0.78 ± 0.08 c	1.64 ± 0.15 b	2.57 ± 0.24 a	0.3
59	Benzyl alcohol	100-51-6	1888	-	MS,RI	0	0.23 ± 0.05 a	0.23 ± 0.03 a	0.25 ± 0.02 a	0.22 ± 0.03 a	0.27 ± 0.04 a	0.23 ± 0.03 a	5.5
60	(R)-(-)-3-Methyl-2-butanol	1572-93-6	1982	-	MS,RI	0	2.27 ± 0.24 b	2.58 ± 0.31 ab	2.68 ± 0.26 a	2.74 ± 0.27 a	2.56 ± 0.18 ab	2.10 ± 0.12 b	/
acids													
61	Acetic acid	64-19-7	1459	1442	MS,RI	581.59 ± 2.24 c	509.50 ± 2.25 a	456.68 ± 3.57 b	451.40 ± 4.28 b	414.41 ± 4.35 c	401.04 ± 2.15 d	254.97 ± 1.57 e	0.1
62	Propanoic acid	79-09-4	1550	1531	MS,RI	11.51 ± 0.25 d	14.25 ± 0.32 a	12.54 ± 0.18 b	12.06 ± 0.15 c	11.21 ± 0.21 de	11.14 ± 0.17 e	10.60 ± 0.27 f	3
63	Butanoic acid	107-92-6	1639	-	MS,RI	31.77 ± 0.25	26.08 ± 0.32	24.93 ± 0.21	21.91 ± 0.18	20.40 ± 0.12 d	17.73 ± 0.08	16.04 ± 0.11 g	0.015
64	Isovaleric acid	503-74-2	1680	1626	MS,RI	24.87 ± 0.52 a	22.85 ± 0.47 b	17.17 ± 0.32 c	16.83 ± 0.25 cd	14.55 ± 0.42 d	13.47 ± 0.38 e	8.36 ± 0.51 f	0.1
65	Pentanoic acid	109-52-4	1749	1714	MS,RI	10.51 ± 0.25 c	12.81 ± 0.35 a	11.04 ± 0.19 b	10.80 ± 0.42 bc	8.91 ± 0.32 d	8.34 ± 0.26 d	6.10 ± 0.21 e	0.5
66	Isocrotonic acid	503-64-0	1791	-	MS,RI	1.86 ± 0.12 d	2.80 ± 0.08 a	2.39 ± 0.07 b	2.43 ± 0.11 b	2.33 ± 0.08 b	2.22 ± 0.07 bc	2.15 ± 0.08 c	/
67	Hexanoic acid	142-62-1	1855	1855	MS,RI	55.27 ± 0.85 a	55.64 ± 1.21 a	49.09 ± 0.54 b	41.68 ± 0.36 c	38.78 ± 0.41 d	37.76 ± 0.52 e	26.99 ± 0.44 f	0.2
68	4-Methylvaleric acid	646-07-1	1878	-	MS,RI	0.29 ± 0.02	0	0	0	0	0	0	/
69	trans-3-Hexenoic acid	1577-18-0	1954	1958	MS,RI	0.25 ± 0.03	0	0	0	0	0	0	1.3
70	Heptanoic acid	111-14-8	1960	1948	MS,RI	4.74 ± 0.24 a	4.51 ± 0.32 a	4.30 ± 0.38 ab	3.97 ± 0.42 b	3.92 ± 0.35 b	3.58 ± 0.37 b	2.44 ± 0.21 c	0.1
71	trans-2-Hexenoic acid	1191-04-4	1982	-	MS,RI	1.51 ± 0.11	0	0	0	0	0	0	/
72	Octanoic acid	124-07-2	2066	-	MS,RI	2.94 ± 0.24 a	3.09 ± 0.26 a	2.80 ± 0.13 a	2.73 ± 0.32 ab	2.64 ± 0.21 b	2.36 ± 0.18 b	2.12 ± 0.15 c	0.5
73	trans-2-Undecenoic acid	15790-94-0	2091	-	MS,RI	0.71 ± 0.05 b	0.88 ± 0.06 a	0.85 ± 0.05 a	0.79 ± 0.03 ab	0.77 ± 0.04 b	0.61 ± 0.03 c	0	/
74	Nonanoic acid	112-05-0	2177	2177	MS,RI	0.74 ± 0.07 ab	0.95 ± 0.08 a	0.93 ± 0.06 a	0.86 ± 0.08 a	0.76 ± 0.07 ab	0.70 ± 0.06 b	0.43 ± 0.05 c	0.0035
75	Decanoic acid	334-48-5	2288	-	MS,RI	2.16 ± 0.15	0	0	0	0	0	0	0.5
hydrocarbons													
76	Undecane	1120-21-4	1104	-	MS,RI	0	0	0	0	0	0.88 ± 0.07 b	8.25 ± 0.12 a	/
77	Dodecane	112-40-3	1203	1200	MS,RI	0	14.01 ± 0.12 e	14.63 ± 0.08 d	14.16 ± 0.11 e	15.82 ± 0.12 c	20.71 ± 0.15 b	32.30 ± 0.32 a	/
78	2,5,6-Trimethyloctane	62016-14-2	1209	-	MS,RI	1.61 ± 0.08	0	0	0	0	0	0	/
79	Styrene	100-42-5	1267	1250	MS,RI	0	3.39 ± 0.28 d	4.19 ± 0.23 c	4.29 ± 0.35 bc	4.59 ± 0.24 b	5.08 ± 0.41 a	5.45 ± 0.32 a	0.022
80	Tridecane	629-50-5	1302	1300	MS,RI	0	4.66 ± 0.33 d	6.75 ± 0.41 c	8.13 ± 0.38 b	8.66 ± 0.35 b	8.96 ± 0.54 b	22.73 ± 1.12 a	/
81	Nonane	111-84-2	1310	900	MS,RI	1.61 ± 0.23 e	1.14 ± 0.21 e	4.19 ± 0.52 cd	4.84 ± 0.38 c	3.52 ± 0.32 d	7.99 ± 0.56 b	24.06 ± 1.21 a	/
furans													
82	2-Acetylfuran	1192-62-7	1518	-	MS,RI	0	0.45 ± 0.09 e	1.06 ± 0.12 d	3.21 ± 0.26 c	3.67 ± 0.21 c	22.40 ± 0.12 b	62.50 ± 0.41 a	0.01
83	Furan-2,5-dicarbaldehyde	823-82-5	1999	-	MS,RI	0	0	0	0	0	0	0.77 ± 0.08	100

^A^ The aroma compounds identified on the TG-WAXMS column. ^B^ The retention index of volatile compounds on the TG-WAXMS column. ^C^ The retention index of volatile compounds from published literature and online library. ^D^ RI: retention index; MS: mass spectrometry [26]. ^E^ The aroma compounds not identified in published literature and online library [27]. ^F^ The threshold of volatile compounds in water referred to in the book (Rychlik M, Schieberle P, Grosch W. *Compilation of odor thresholds, odor qualities and retention indices of key food odorants* [M]. Germany: Deutche Forschungsanstalt fur Lebensmittelchemie: Garching, 1998). ^G^ Values with different superscript Roman letters (a–g) in the same row are significantly different according to the Duncan test (*p* < 0.05). ^H^ The threshold of volatile compounds not found. ^I^ (B1) Microwave drying time 0 min; (B2) Microwave drying time 1 min; (B3) Microwave drying time 2 min; (B4) Microwave drying time 2.5 min; (B5) Microwave drying time 3 min; (B6) Microwave drying time 4 min; (B7) Microwave drying time 5 min.

**Table 2 foods-12-03012-t002:** The major odor-active compounds identified by GC-O in seven jujube slices with the odor activity value (OAV) method and detection frequency analysis (DFA).

No	Compound Name	CAS	OAV ^A^							DF ^B^							Odor Quality ^C^
			B1	B2	B3	B4	B5	B6	B7	B1	B2	B3	B4	B5	B6	B7	
aldehydes																	
2	Hexanal	66-25-1	4.3	3.7	2.9	2.6	2.2	<1	<1	1	0 ^D^	0	0	0	0	0	green
5	Nonanal	124-19-6	2.4	2.1	1.5	1.2	0.9	0.6	0.3	1	0	0	0	0	0	0	fatty
7	Furfural	98-01-1	<1	<1	<1	1	2.1	4.9	10.3	0	0	0	0	0	0	3	burnt
10	5-Methyl-2-furanaldehyde	620-02-0	<1	<1	<1	<1	1.07	2.76	4.64	0	0	0	0	1	5	7	burnt
12	5-Hydroxymethyl-2-furaldehyde	67-47-0	<1	<1	<1	<1	<1	<1	<1	0	0	0	0	1	4	6	burnt
esters																	
15	Ethyl hexanoate	123-66-0	278.5	8.1	7	7	4.2	<1	<1	3	2	1	1	0	0	0	fruity
26	γ-Butyrolactone	96-48-0	<1	<1	1.15	1.47	1.21	1.03	<1	5	5	6	6	6	5	5	caramel
27	4-Hexanolide	695-06-7	<1	<1	<1	<1	<1	<1	<1	0	0	0	0	3	6	7	sweet
ketones																	
34	2,3-Butanedione	431-03-8	<1	4	9.7	12.7	7.3	2.3	<1	9	10	10	10	10	10	10	roasted sweet
35	Acetoin	513-86-0	<1	1.5	2.6	3.2	2.2	1	<1	10	10	10	10	10	10	10	roasted sweet
38	4-Cyclopentene-1,3-dione	930-60-9	<1	<1	<1	1.42	1.03	<1	<1	0	0	2	6	3	2	2	caramel
40	5-Methyl-2(5H)-furanone	591-11-7	0	0	0.005	0.49	0.63	3.5	7.51	0	0	0	0	3	6	9	bitter
45	4-Hydroxy-2,5-dimethyl-3(2H)furanone	3658-77-3	<1	<1	<1	1.75	<1	<1	<1	0	0	1	3	2	1	1	caramel
49	2,3-Dihydro-3,5-dihydroxy-6-methyl-4(H)-pyran-4-one	28564-83-2	0	0.06	0.3	0.58	0.83	2.65	5.34	0	0	1	3	7	8	8	burnt, bitter
50	3,5-Dihydroxy-2-methyl-4-pyrone	1073-96-7	<1	<1	<1	<1	<1	<1	1.31	0	0	0	0	1	5	6	burnt, bitter
alcohols																	
52	1-Octen-3-ol	3391-86-4	5.7	4.9	4.9	4.7	4.5	4.2	<1	8	7	7	7	7	7	6	mushroom
57	2-Furanmethanol	98-00-0	<1	<1	<1	<1	<1	<1	<1	0	0	0	0	1	3	5	burnt
acids																	
61	Acetic acid	64-19-7	5.8	5.1	4.6	4.5	4.1	4	2.5	10	10	10	10	10	10	10	sour
63	Butanoic acid	107-92-6	2.1	1.7	1.7	1.5	1.4	1.2	1.1	3	3	3	3	3	3	4	smelly
67	Hexanoic acid	142-62-1	<1	<1	<1	<1	<1	<1	<1	10	10	10	10	10	10	8	smelly
furans																	
82	2-Acetylfuran	1192-62-7	<1	<1	<1	<1	<1	2.24	6.25	0	0	0	0	1	6	7	bitter

^A^ Odor activity value (OAV) ratio of concentration to odor threshold. ^B^ The number of people who recognize the odor quality through GC-O (total panelists, *n* = 10). ^C^ Odor quality perceived by GC-O analysis. ^D^ Not perceived.

## Data Availability

The data used to support the findings of this study can be made available by the corresponding author upon request.

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
