# Peer review of "Profiling the Major Aroma-Active Compounds of Microwave-Dried Jujube Slices through Molecular Sensory Science Approaches"

_foods, 2023, doi:10.3390/foods12163012_

Round 1

Reviewer 1 Report

General comment

The manuscript ''Profiling the major aroma-active compounds of microwave-Dried jujube slices by molecular sensory science approache'' described the aroma-active compounds in dired jujube slices by microwave-dried treatments. In total, 21 major aromatic active compounds were detected. The correlation between aromatic compounds and specific taste attributes was also done.

Article should be corrected from several aspects. At first, there are too many language mistakes and manuscript should be checked by professional linguistic or native speaker.

Authors should check names of all compounds. Additionally, all compounds should be written on the same way.

Table 1 should be put in Landscape orientation to be easy to folow presented data. Also, it is not necessary to repat the names of column on each new page.

 Minor comments

Title

Line 2: change ''Dried'' into ''dried''

Abstract

Line 11: change ''dired'' into ''dried'' (and through all text)

Line 14: add space before ''were''

Line 17: omit one dot after ''jujube''

Line 28: omit space after ''regression''

1. Introduction

Line 32: add in brackets '', family Rhamnaceae''

Lines 33, 36, 37, and through all text: add space before bracket with reference number

Lines 50, 51: change ''jujube'' into ''jujuba'' and put ''Ziziphus jujuba'' into Italic

Line 57: add space before ''It''

Line 74: explain abbreviations ''OAV'' and ''DFA''

Line 76: explain abbreviations ''PLSR''. It should be also explained here, not only in Abstract

Line 76: add space before ''The''

2. Materials and Methods

Line 91: add space before ''kg''

Line 92: change ''china'' into ''China''

Line 99: change ''jujube'' into ''Jujube''

Lines 99, 100: add space before ''g''

Line 107: explain abbreviations ''DVB/CAR/PDMS''

Lines 116, 149: add town after ''Thermo''

Line 139: add space before ''mo'' and put ''zero or letter'' (it is not clear) into subscript

Line 143: explain abbreviations ''OAV'' in line 74 (not here)

Line 144: add space before ''1''

Lines 174: add town before ''Canada''

Lines 189–191: omit these sentences

3. Results and Discussion

Lines 200–200: rephrase this sentance

Line 205: add ''were'' after ''contains''

Lines 213, 217: change web address with reference number (from section References)

Line 221: change wavy hyphnen with normal hyphen

Lines 225, 230, 231: add space before brackets

Line 258: what is ''jujube-ID''?

Line 265: ''Furanone'' or ''furanone''? See lines 263, 264, and 293.

Line 289: ''Hexanoic'' or ''hexanoic''?

Line 293: 5-ethyldihydro-(Sweet)?

Line 308: omit space before comma

Line 314: 2,5-dimethyl-4-hydroxy-3(2H)-furanone(caramel)?

Line 327: change ''2'' into ''two''

Line 329: 2-Furanmethano?

Line 342: change ''hierarchical'' into ''Hierarchical'' and omit one ''in''

Lines 342–343: add space before brackets

Table 2 should be wider. Please, change.

Line 354: Figure.3.?

 References

References should be presented according to journal style. So, check them all and make corrections. For example, the style of writting journal names should be the same.

Lines 462–463, 470, 474: put ''Ziziphus jujuba'' into Italic

Line 500: omit one dot before ''Sensory''

Lines 511–512: put ''Solanum quitoense'' into Italic

Line 513: put ''Agaricus bisporus'' into Italic

Author Contributions

Lines 445–450: use the same style through all paragraph

Funding

Line 451: Please add:??

I am not native speaker of English Language but I was detected countless mistakes. Manuscript should be checked by professional linguistic or native speaker.

Author Response

The manuscript ''Profiling the major aroma-active compounds of microwave-Dried jujube slices by molecular sensory science approache'' described the aroma-active compounds in dired jujube slices by microwave-dried treatments. In total, 21 major aromatic active compounds were detected. The correlation between aromatic compounds and specific taste attributes was also done.

Article should be corrected from several aspects. At first, there are too many language mistakes and manuscript should be checked by professional linguistic or native speaker.

Modify reply:Thank you very much for your recognition of our work. We fully acknowledge the questions you raised, and have made modifications and replied to all questions.

Authors should check names of all compounds. Additionally, all compounds should be written on the same way.

Modify reply:All compound names have been checked and standardized in writing. The modified sections have been marked in red font in the document.

Table 1 should be put in Landscape orientation to be easy to folow presented data. Also, it is not necessary to repat the names of column on each new page.

Modify reply:Table 1 has been modified by removing the page titles and streamlining the content within the table.

 Minor comments

Modify reply:Thank you to the reviewer for carefully examining the manuscript and providing detailed suggestions for revision. We have addressed each of the reviewer's requests regarding writing, formatting, and other issues, and have indicated the changes in red font within the manuscript.

Title

Line 2: change ''Dried'' into ''dried''

Has been modified

Abstract

Line 11: change ''dired'' into ''dried'' (and through all text)

Has been modified

Line 14: add space before ''were''

Has been modified

Line 17: omit one dot after ''jujube''

Has been modified

Line 28: omit space after ''regression''

Has been modified

  1. Introduction

Line 32: add in brackets '', family Rhamnaceae''

Has been modified

Lines 33, 36, 37, and through all text: add space before bracket with reference number

Has been modified

Lines 50, 51: change ''jujube'' into ''jujuba'' and put ''Ziziphus jujuba'' into Italic

Has been modified

Line 57: add space before ''It''

Has been modified

Line 74: explain abbreviations ''OAV'' and ''DFA''

Has been modified

Line 76: explain abbreviations ''PLSR''. It should be also explained here, not only in Abstract

Has been modified

Line 76: add space before ''The''

Has been modified

  1. Materials and Methods

Line 91: add space before ''kg''

Has been modified

Line 92: change ''china'' into ''China''

Has been modified

Line 99: change ''jujube'' into ''Jujube''

Has been modified

Lines 99, 100: add space before ''g''

Has been modified

Line 107: explain abbreviations ''DVB/CAR/PDMS''

Has been modified

Lines 116, 149: add town after ''Thermo''

Has been modified

Line 139: add space before ''mo'' and put ''zero or letter'' (it is not clear) into subscript

Has been modified

Line 143: explain abbreviations ''OAV'' in line 74 (not here)

Has been modified

Line 144: add space before ''1''

Has been modified

Lines 174: add town before ''Canada''

Has been modified

Lines 189–191: omit these sentences

Has been modified

  1. Results and Discussion

Lines 200–200: rephrase this sentance

Has been modified

Compared to reference [18], some new compounds were found, namely, 5-ethyldihydro-2(3H)-furanone,4-cyclopentene-1,3-dione, 5-methyl-2(5H)-furanone, 2,5-dimethyl-4-hydroxy-3(2H)-furanone, and 3-hydroxy-2-methyl-4H-pyran-4-one.

Line 205: add ''were'' after ''contains''

Has been modified

Lines 213, 217: change web address with reference number (from section References)

Has been modified

Line 221: change wavy hyphnen with normal hyphen

Has been modified

Lines 225, 230, 231: add space before brackets

Has been modified

Line 258: what is ''jujube-ID''?

“Jujube ID” is the characteristic aroma of dried jujube, as explained in section 3.3.

Line 265: ''Furanone'' or ''furanone''? See lines 263, 264, and 293.

Has been modified

Line 289: ''Hexanoic'' or ''hexanoic''?

Has been modified

Line 293: 5-ethyldihydro-(Sweet)?

Has been modified

Line 308: omit space before comma

Has been modified

Line 314: 2,5-dimethyl-4-hydroxy-3(2H)-furanone(caramel)?

Has been modified

Line 327: change ''2'' into ''two''

Has been modified

Line 329: 2-Furanmethano?

Has been modified

Line 342: change ''hierarchical'' into ''Hierarchical'' and omit one ''in''

Has been modified

Lines 342–343: add space before brackets

Has been modified

Table 2 should be wider. Please, change.

Has been modified

Line 354: Figure.3.?

Has been modified

 References

References should be presented according to journal style. So, check them all and make corrections. For example, the style of writting journal names should be the same.

The references have been reviewed and revised.

Lines 462–463, 470, 474: put ''Ziziphus jujuba'' into Italic

Has been modified

Line 500: omit one dot before ''Sensory''

Has been modified

Lines 511–512: put ''Solanum quitoense'' into Italic

Has been modified

Line 513: put ''Agaricus bisporus'' into Italic

Has been modified

Author Contributions

Lines 445–450: use the same style through all paragraph

Has been modified

Funding

Line 451: Please add:??

Modify reply:Already removed.

Reviewer 2 Report

This manuscript deals with aroma-active compounds of microwave-dried jujube slices. The study includes the identification of volatile compounds, the search for odorants by GCO and a sensory profile QDA. Are there different jujube cultivars that might have different aromatic characteristics? If so, the choice should be justified in the introduction.

The study is rather well done and the manuscript rather clear.

Comments and questions:

How long is a GCO sequence for each panellist?

L154: What did the panellists' training consist of? How their performances were checked?

L156: How are the qualitative odour statements made by panellists recorded?

L158: aroma profile

L160-167: Why not have the panellists generate their own terms?

10 judges: it is rather low for QDA.

L172: How were the panellists trained? Over how many sessions?

L173: did they rinse their mouth during the 3 min intervals?

Table 1 would be much easier to read in landscape mode, at least to avoid having the IRs on 2 lines. as presented, it is virtually unreadable. It is the same for Table 2.

Table 2 and L 354: For sour, sweet, acid and bitter “odors”, is it taste-associated odors?

What are the prospects for this work? To be completed in the conclusion.

Author Response

This manuscript deals with aroma-active compounds of microwave-dried jujube slices. The study includes the identification of volatile compounds, the search for odorants by GCO and a sensory profile QDA. Are there different jujube cultivars that might have different aromatic characteristics? If so, the choice should be justified in the introduction.

Modify reply:

Different cultivars of jujube can indeed exhibit variations in flavor.The Grey jujube is known for its high yield and is a primary raw material for dried jujube slices. So,we selected the Grey jujube as experimental sample. And the main purpose of this article is to discriminate the aroma-active compounds in dired jujube slices by microwave-Dried treatments and understand their sensory attributes.

The study is rather well done and the manuscript rather clear.

Comments and questions:

How long is a GCO sequence for each panellist?

Modify reply:

The whole time of GC-MS analysis was 48 min, in which 30 min for GC-O sequence. Before the GC-O frequency analysis , the retention time of the first compound that can be smelled was confirmed by pre-experiment. We found that the retention time of the first compound was10.07min, so the start time of the GC-O sequence for each panellist was 9 min.

On the other hand, long time sniffing could cause olfactory fatigue and based on the results of the preliminary experiment, we determined the end time to be 39 minutes.

trained panelists

L154: What did the panellists' training consist of? How their performances were checked?

Modify reply:

We have supplemented the content related to the training process.

Original Sentence:A group of 10 trained sensory panelists, consisting of five males and five females, were employed to determine the detection frequency (DF) of the samples. The panelists recorded the retention time and odor quality during substance detection.

Revised Version:Ten assessors, consisting of 5 males and 5 females, with over 300 hours of technical experience were selected. Prior to olfactory analysis, artificial odor solutions were used for odor identification. Assessors were instructed to record the retention time and the detection frequency (DF) of the samples. 

L156: How are the qualitative odour statements made by panellists recorded?

Modify reply:

The study employed the detection frequency analysis method, and the experimenters only recorded the retention time and the detection frequency (DF) of the samples.

L158: aroma profile

Modify reply:Vocabulary spelling errors have been corrected.

L160-167: Why not have the panellists generate their own terms?

Modify reply:

Thank you for the reviewer's comments. The sensory descriptors used were proposed by members of the sensory evaluation team according to relevant references, and words with a voting rate of 80% or above were considered as evaluation words. This has been introduced in the manuscript.

10 judges: it is rather low for QDA.

Modify reply:The number of evaluators was determined based on relevant literature, as shown below.

Tobin, R., Moane, S., & Larkin, T. Sensory evaluation of organic and conventional fruits and vegetables available to Irish consumers. International journal of food science & technology, 2013,48(1), 157-162.

Bi, S., Xu, X., Luo, D., Lao, F., Pang, X., Shen, Q., ... & Wu, J. Characterization of key aroma compounds in raw and roasted peas (Pisum sativum L.) by application of instrumental and sensory techniques. Journal of Agricultural and Food Chemistry, 2020, 68(9), 2718-2727.

L172: How were the panellists trained? Over how many sessions?

Modify reply:

The training primarily focused on the principles of sensory analysis and quantitative descriptive analysis, following the ISO international standard (8586-1, 1993). During the three-month training period, which consisted of 2 hours per week, the panelists were trained to develop the ability to differentiate between different levels of aromas. The subsequent stage of training involved describing and discussing the aroma characteristics of dried jujube slices. Modified and annotated in the manuscript.

L173: did they rinse their mouth during the 3 min intervals?

Modify reply:

Dear Reviewer, the main focus of the experiment was on aroma evaluation and not taste evaluation. Therefore, only aroma training was conducted, and no mouth rinsing was performed.

Table 1 would be much easier to read in landscape mode, at least to avoid having the IRs on 2 lines. as presented, it is virtually unreadable. It is the same for Table 2.

Modify reply:

The table has been reviewed, revised, and improved.

Table 2 and L 354: For sour, sweet, acid and bitter “odors”, is it taste-associated odors?

Modify reply:

Sour, sweet, acid and bitter “odors” were not taste-associated odors. Lina Wang used“roast”, “sweet”, and “sour” to describe the ordor of jujubes, besides, Francisca Hernández used bitterness as an important attribute to analyse the flavour of Spanish jujube (Ziziphus jujuba Mill.) fruits.We also added these two research in this paper.

Revised Version:Lina Wang [30] used“roast,” “sweet,” “green,” “sour,” “fruity,” and “mellow” attributes to describe the odor of jujubes. Moreover, bitterness and jujube ID were evaluated as important attributes to analyze the flavor of Spanish jujube (Ziziphus jujuba Mill.) fruits [31]. Notably, different substances are present in different odors. A total of 15 aromatic aroma compounds were identified by matching LRI and mass spectrum. These compounds were then categorized into 5 groups according to their odor characteristics. They were roasted sweet odorants (34, 35 in Table 1), caramel-like odorants (26, 38 in Table 1), burnt-like odorants (10, 12, 49, 50, and 57 in Table 1), bitter-like odorants (40,49,50 and 82 in T and Table 1), and jujube-ID odorants (13, 27, 52, and 67 in Table 1). The jujube-ID odorants consisted of mushroom, smelly, sour, and sweet substances.

What are the prospects for this work? To be completed in the conclusion.(

Modify reply:

According to the reviewer's request, the manuscript has been updated with a description of the significance and prospects of the work.

Revised Version:

These conclusion would contribute to the understanding of the material basis of dired jujube slices, aroma and their change regularities and provide a theoretical basis for the improvement of aroma quality and the regulation of peculiar flavor to produce high-quality jujube slices.

Reviewer 3 Report

The authors performed an experiment where they analyzed the effect of microwave heating at different time exposures on the profile of volatile compounds in jujube slices. Meanwhile, the authors set out as a aim of article to determine the most important volatile compounds of jujube. The authors boast in their conclusions which compounds were first time detected and which have high OAV and DF, but they completely ignore in which samples these compounds were detected (whether in the unheated sample or after microwave treatment). The authors simplify the evaluation of their findings by not separating them into the volatile compound profile of untreated jujube slices and the changes in the volatile compound profile under extended microwave exposure. This is especially evident in the conclusion section.Therefore, I believe that the entire article needs to be revised and systematized before any review can take place.

Another thing is also that I get the impression that the authors did not read the text they submitted to the journal at all - it is full of linguistic errors, stylistic errors, strange text, etc.

line 27- dired or dried ? – please check here and in entire article

line 34-35- this is data form 2018, but now we have 2023 - please update the harvesting area and production volume of jujube

why authors write dried jujube with capital letters across entire article?

line 72-80- The authors provided the same text twice regarding the purpose and scope of the work.

line 88-89 - please specify what the transport conditions were. Water content and sugar content were measured before or after transport? This is relevant to the research experiment.

line 98- In what were the powdered fruits packed and how were they stored for the tests?

line 100 – why authors add to puree 1% sodium fluoride solution? please justify in the text.

line 102- “….and then the puree sample was immediately used for the next experiment.” – please change on  “….and then the puree samples were immediately used for the analytical determinations with GC usage.”

line 107- Please justify the choice of fiber for the analyses.

line 113- Please justify the choice of volatiles extraction parameters.

line 125- what the authors have done is a semiquantitative analysis of volatile compounds. A full quantitative analysis would have been done by the authors if they had standards of all the identified compounds. So please use the phrase semiquantitative determination of volatile compounds in the article.

line 135-136- on what basis did the authors decide to take calibration factor 1.00 for the calculations?

line 142- please describe more how the authors calculated the OAVs of identified volatiles.

line 185-187- Please describe the PLSR analysis in more detail and elaborate on the acronym of the name. Are all volatile compounds qualified for this analysis? If yes/no, please write on what basis?

line 188- change name on Results and Discussion

line 189-191-This is a fragment from the mdpi format, please remove it.

Table 1 - The table is very unreadable, poorly formatted. The authors want to convey too much information in one table. I suggest that another table be created from some of the columns and placed in the supplementary material.

Table 2- The quality of this table is better than the first one, but it still needs to improve the formatting and suggests subtracting some information to make it more readable and informative for the reader.

The authors report all their results supporting themselves with a few articles. A real discussion of the results with the literature is missing.

Entire text should be checked in terms of language and formatting as it is full of linguistic errors, stylistic errors, strange text, etc.

Reviewer 4 Report

The originality of the study and the novelty it brings to the field is of actuality. The purpose of the article and its significance is stated clearly. The paper is well-structured, the abstract is concise, and the topic; in the introduction is supported by well-selected bibliographic data. The Experimental and Modeling Approach correctly.

It is important to state clearly the implications for research, practice, and society.

Avoid repetition and try to focus on the main ideas regarding the results. At the same time, the Results and Discussions section could be improved by studying other papers in the field.

Round 2

Reviewer 1 Report

As I can see from provided text the manuscript did not carefully checked neither by authors nor by professional linguistic. I detected again more or less the same mistakes (for example, see lines 26, 43, 44, 80, 95, 102). Also, authors did not make some corrections although they said that they done it (for example, see lines 26 and 44).

Minor comments

Abstract

Line 9: add space after ''4-hexanolide,''

Lines 10 and 15: ''-2.5-dimethyl-'' or ''-2.5-dimethyl-''

 1. Introduction

Line 26: add in brackets'', family Rhamnaceae''

Lines 44: change ''jujube'' into ''jujuba''

2. Materials and Methods

Line 127: put ''s'' (in ''ms'') into subscript

3. Results

Line 186: change ''dried'' into ''Dried''

Line 196: add space after ''acid,''

Table 1: some values were put in Bold. Explain way below the table.

Lines 214, 215: add space before ''Microwave''

Lines 217, 218: add space before ''drying'' and ''min''

Table 2: column B3 and the last column should be wider

Line 302: add space before bracket

Lines 326, 327, 355, 356: add space before bracket

4. Conclusions

Line 357: change ''4. Results and Discussion'' into ''4. Conclusions''

References list is missing

I am not native speaker, but I noticed again mistakes in English Language.

Reviewer 3 Report

Please see the detailed comments attached.
